# To What Extent Should We Rely on Antibiotics to Reduce High Gonococcal Prevalence? Historical Insights from Mass-Meningococcal Campaigns

**DOI:** 10.3390/pathogens9020134

**Published:** 2020-02-18

**Authors:** Chris Kenyon

**Affiliations:** 1HIV/STI Unit, Institute of Tropical Medicine, 2000 Antwerp, Belgium; kenyon@itg.be; Tel.: +32-3-2480796; Fax: +32-3-2480831; 2Division of Infectious Diseases and HIV Medicine, University of Cape Town, Anzio Road, Observatory 7700, South Africa

**Keywords:** *Neisseria gonorrhoeae*, AMR, *Neisseria meningitides*, commensal Neisseria

## Abstract

In the absence of a vaccine, current antibiotic-dependent efforts to reduce the prevalence of *Neisseria gonorrhoeae* in high prevalence populations have been shown to result in extremely high levels of antibiotic consumption. No randomized controlled trials have been conducted to validate this strategy and an important concern of this approach is that it may induce antimicrobial resistance. To contribute to this debate, we assessed if mass treatment in the related species, *Neisseria meningitidis,* was associated with the emergence of antimicrobial resistance. To this end, we conducted a historical review of the effect of mass meningococcal treatment programmes on the prevalence of *N. meningitidis* and the emergence of antimicrobial resistance. We found evidence that mass treatment programmes were associated with the emergence of antimicrobial resistance.

## 1. Introduction

The World Health Organization’s plan to reduce the incidence of *Neisseria (N.) gonorrhoeae* by 90% by 2030 faces two growing challenges—antimicrobial resistance and rising rather than falling incidence of *N. gonorrhoeae* in many key populations [1,2]. A number of the strategies advocated to reduce gonococcal incidence such as intensified screening, partner tracing/expedited partner therapy and doxycycline pre-exposure prophylaxis, depend on increasing antibiotic consumption [2,3]. These increases can be large. Screening for gonorrhoea/chlamydia at three sites every three months in HIV pre-exposure prophylaxis (PrEP) cohorts, for example, has been shown to result in very large macrolide and cephalosporin exposures. Macrolide exposure, for example, can reach 4400 defined daily doses/1000 population per year, which is approximately 20 times the population consumption of a country such as Sweden [4]. A concern of such high levels of antibiotic consumption is the induction of antimicrobial resistance (AMR) in *N. gonorrhoeae* and other organisms [4]. 

To assist in the evaluation of this concern, we undertook a historical review of the effect of mass antimicrobial treatments on antimicrobial susceptibility of the related *Neisseria*, *N. meningitidis.* There have been few mass treatment trials of *N. gonorrhoeae.* These studies have typically found that mass treatment has no effect [5,6,7], or only a temporary effect on gonorrhoea incidence/prevalence [8,9]. Only one of these studies evaluated the effect on AMR. Although this study found a temporal association between mass treatment and the emergence of gonococcal AMR, its contemporary relevance is reduced by the fact that it was conducted using penicillin in the 1960s [8,9]. 

Considerably more mass treatment studies have been conducted for *N. meningitidis*. These mass treatment studies involved the widespread administration of antibiotic therapy (chemoprophylaxis) to a community with either excess cases of meningococcal disease or a raised prevalence of asymptomatic *N. meningitidis* [10,11,12,13]. 

Although there are important differences in the mode of transmission, preferred site of colonization, clinical presentation and host immune response between *N. meningitidis* and *N. gonorrhoeae*, there are also considerable similarities [14,15]. Despite being the only two species in the *Neisseria* genus that are classified as strict human pathogens, the majority of both infections are asymptomatic and resolve spontaneously. Both infections cluster in particular population groups. In the case of meningococcus, and in keeping with its respiratory transmission, epidemics and high carriage rates are predominantly associated with young adults living in crowded conditions [11,12,13]. *N. gonorrhoeae* is sexually transmitted and thus a high prevalence has been linked to factors such as high rates of sexual partner turnover which generate dense sexual networks and high equilibrium prevalences of *N. gonorrhoeae* [2,16,17,18,19]. In the case of PrEP cohorts, for example, modelling studies suggest that the five to ten sexual partners per three months reported by PrEP recipients generate the high prevalence of *N. gonorrhoeae* in these populations—typically around 10% [16,20]. Crucially, the two infections are genotypically closely related and able to exchange DNA between one another and commensal *Neisseria* via well-developed systems of transformation [21,22,23]. The uptake of DNA from other *Neisseria* species has been established as a key way that both the gono- and meningococcus have acquired antimicrobial resistance [21,22,23]. *N. gonorrhoeae* has been noted to be more susceptible to the emergence of AMR than *N. meningitidis* [24]. These considerations suggest that if mass treatment of *N. meningitidis* is associated with the emergence of AMR, this would provide a cautionary warning for using antibiotic based strategies to reduce the prevalence of *N. gonorrhoeae* in high prevalence settings such as PrEP cohorts.

## 2. Effect of Mass Treatment on Prevalence of *N. meningitidis*, Meningitis Cases and AMR

A recent review paper by MacNamara et al. evaluated the effect of mass treatment of *N. meningitidis* on the prevalence of the bacteria and the emergence of AMR in over 33 studies [10]. The authors concluded that the intervention was highly effective in reducing cases of meningitis and, when an effective antibiotic was used at over 75% population coverage, this resulted in a 50% to 80% reduction in carriage in the short term (median follow up six weeks). In the one study with less than 75% coverage, there was no reduction in carriage [25]. This review paper did not evaluate the long-term effects. One of the few studies to assess this was a study from a Kibbutz, in Israel, that found that mass treatment resulted in a decline in carriage but this effect only lasted six months [26]. 

Although the effect on AMR was not assessed in all studies, when it was assessed, AMR emerged fairly frequently. Resistance to rifampicin was particularly evident and found in all three community studies where this was assessed [10,25,27,28]. Rifampicin resistance was also noted in cases following two mass therapy interventions in the United States of America (US) military [10]. Sulfadiazine was used extensively in the US military to prevent meningococcal disease from the 1940s to the 1960s [29]. This widespread use was thought to play a role in the rapid and extensive emergence of AMR in the 1950s and 1960s [29]. Only one study tested for ciprofloxacin resistance following use of this agent. This study found no ciprofloxacin resistance but only evaluated for resistance six months after the intervention [30]. No studies evaluated the emergence of resistance to other antimicrobials such as ceftriaxone and azithromycin. 

### 2.1. Individual Level Assessment

A systematic review of the efficacy of various antibiotics for the eradication of *N. meningitidis* carriage found that penicillin, rifampicin, minocycline, ciprofloxacin and ceftriaxone were effective at eradicating carriage for up to four weeks [31]. Eleven trials reported the susceptibility of persistent isolates to the antibiotic used for elimination. Six of these studies evaluated the induction of AMR by rifampicin. Resistance was found in persistent isolates in three of these six studies—the prevalence of resistance was between 10% and 27% [31]. The use of other antibiotics was not associated with the selection of resistance.

### 2.2. Association between Overcrowding and N. meningitidis Prevalence/outbreaks

We could not find any systematic reviews on this topic, but there was broad consensus in the literature we reviewed that overcrowding (particularly for young adults) played a crucial role in outbreaks of meningococcal disease and increases in prevalence [11,13,26,32]. Glover was the first to describe this association in 1917 in an outbreak of meningococcal disease in soldiers in military recruitment camps. Using nasopharyngeal cultures to evaluate meningococcal colonization prevalence, he noted a steep increase in prevalence following the overcrowding of recruits (Figure 1) [13]. The camp was designed to accommodate 800 men but was accommodating close to 6000 men by the start of the epidemic. Of note, meningococcal prevalence decreased following measures that included reducing overcrowding (Figure 1). A range of subsequent studies and reviews of the topic have produced similar findings [11,26,32].

## 3. Discussion

Mass treatment was fairly effective in the short term in reducing the prevalence of *N. meningitidis* but this effect did not appear to persist beyond six months. Mass treatments appeared to result in the emergence of AMR to rifampicin and sulphadiazine. There was little or no data for other classes of antibiotics. 

The utility of these findings is limited by the fact that the effect of mass treatment with the antibiotics currently mostly used to treat *N. gonorrhoeae* (azithromycin/ceftriaxone) was not assessed. There are also important biological differences between *N. meningitidis* and *N. gonorrhoeae*, as well as differences between the mass administration of antibiotics during a meningococcal outbreak and the sustained high levels of antibiotic exposure in a PrEP cohort.

Despite these important reservations, the fact that AMR can emerge so rapidly in the related *N. meningitidis* does provide additional motivation to be alert for the emergence of gonococcal AMR in PrEP and other high antibiotic exposure populations. There is increasing evidence that horizontal gene transfer plays an important role in the genesis of AMR in *N. meningitidis* and even more so in *N. gonorrhoeae.* For example, it has been established that transformation from commensal pharyngeal *Neisseria* spp. played an important role in *N. gonorrhoeae’s* acquisition of resistance to extended spectrum cephalosporins [33,34,35]. The acquisition of AMR via commensals can operate over much longer periods than direct selection during antibiotic therapy, as the commensals (and their resistance conferring genes) persist for longer periods than *N. gonorrhoeae*. These resistance genes can then be taken up months later by incoming gono- and meningococci [21,22,23]. We could not find any studies that evaluated the effect of mass treatments on the antibiotic susceptibility of commensal *Neisseria* species and, thus, we were unable to evaluate this effect. Unsurprisingly, however, studies have found a link between antibiotic susceptibility of commensal *Neisseria* and antibiotic consumption [36]. A study from Japan found high proportions of circulating *Neisseria subflava* to have high miminum inhibitory concentrations for penicillin, cefixime, ciprofloxacin and tetracycline [36]. This was thought to be related to the high levels of the corresponding antimicrobial consumed in Japan [36,37]. More direct evidence of this association comes from a study from Vietnam which found decreased cephalosporin susceptibility in commensal *Neisseria* to be related to recent cephalosporin consumption [38].

Two trials have been conducted to assess if doxycycline pre- and post-exposure could reduce the incidence of bacterial STIs including gonorrhoea [39,40]. Both studies found evidence of moderate reductions in chlamydia and syphilis incidence, but not gonorrhoea. The effect on AMR of pathogens and commensal organisms was not assessed in these studies.

Most people are persistently colonised during their lifetime with a variety of commensal *Neisseria* species, any of which can become a reservoir for AMR upon repeated exposure to antibiotic treatment of the host [41]. As a result, the selection pressure imposed by high antibiotic consumption is likely to be seen in these commensals before it becomes evident in gono- and menginococci [14]. As a result, commensal *Neisseria* could serve as an AMR early warning sign and it may be prudent to monitor the antibiotic susceptibilities of these commensal *Neisseria* species in high gonococcal prevalence and high antibiotic consumption populations, such as those on PrEP [42,43]. 

A further relevant parallel between gono- and meningococci is how the prevalence of both infections is strongly influenced by underlying dense contact networks—sexual networks and spatial networks, respectively [17,18,19]. It is these underlying networks which are thus primary determinants of high prevalence and should be the targets of radical prevention [19,43]. The high rates of partner change reported by PrEP recipients, for example, are responsible for the high prevalence of *N. gonorrhoeae* in this group [16,20]. This high network connectivity could be reduced by increased condom usage or reduced rates of partner turnover. Vaccination represents an enticing alternative strategy—as demonstrated by the efficacy of vaccination against *N. meningiditis* [44,45,46,47,48,49]. Although progress has been made in the development of a gonococcal vaccine, the best available vaccine (*N. meningitidis* group B outer membrane vaccine), appears to only have limited efficacy and for a short period [50,51,52]. In the absence of an effective vaccine, it is understandable that efforts to control increasing incidence of *N. gonorrhoeae* have focused on strategies relying on antibiotics. The evidence reviewed here suggests that extensive use of antibiotics to control *N. meningitidis* prevalence runs the risk of inducing AMR. These findings provide further justification to reconsider antibiotic based strategies to reduce gonococcal prevalence—such as three-monthly screening for gohorrhoea/chlamydia in PrEP cohorts. They also provide further motivation for enhanced surveillance of AMR in all *Neisseria* species in high prevalence, high antibiotic consumption populations.

## Figures and Tables

**Figure 1 pathogens-09-00134-f001:**
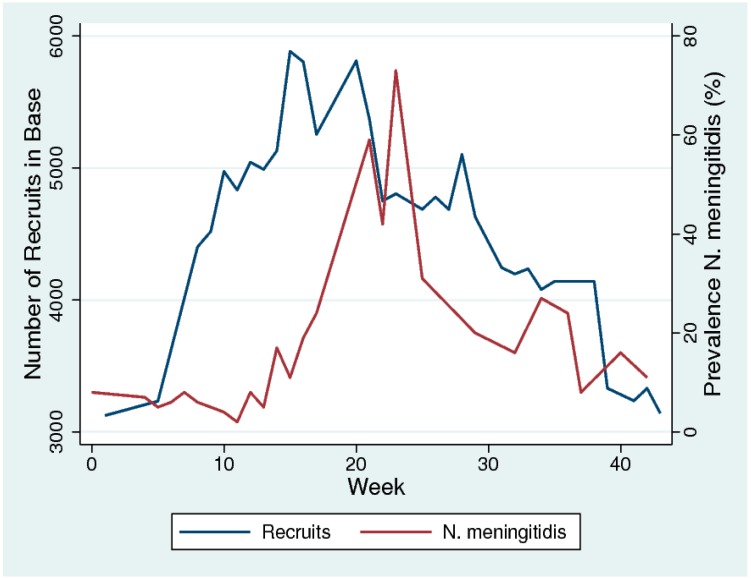
The temporal association between increased overcrowding (number of recruits) and prevalence of *N. meningitidis* in military recruits in a training camp in the south of England in 1917. Week 1 represents the first week of September 2017. (Based on data from [10] digitized with WebPlotDigitizer-4.2 and figure made in Stata 16.0).

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
