# Peer review of "To What Extent Should We Rely on Antibiotics to Reduce High Gonococcal Prevalence? Historical Insights from Mass-Meningococcal Campaigns"

_pathogens, 2020, doi:10.3390/pathogens9020134_

Round 1

Reviewer 1 Report

To the Author

This is a well written article. There were only two small issues about terminology that I needed to raise.

First, I am not sure that the term "commensal Neisseriae"is correct. Scientific usage for the plural here is "commensal Neisseria species (spp.)""

Line 128. In terms of the prevalence of commensal Neisseria species in the community, their prevalence is not 100%. Knapp (Clin Micro Reviews 1988, 1:415) published an excellent review about them as they are not commonly researched. Based on his paper page 422, I would favour the phrase"Most people are persistently colonised during their lifetime with a variety of commensal Neisseria spp. any of which can become a reserviour for AMR upon repeated exposure to antibiotic treatment of the host".

I hope this helps.

Author Response

Thank you for the useful suggestions.

I have changed all references to "Neisseriae"along the lines suggested.

The sentence kindly suggested by the the reviewer has been used and the Knapp reference added.

Reviewer 2 Report

1) Line -22 Please add (N.) next to Neisseria the first time you use it

2) General comments to references:

Some recent papers regarding the effect of meningococcal vaccines on carriage should be added, such as:

Meningococcal B Vaccine and Meningococcal Carriage in Adolescents in Australia.

Marshall HS, McMillan M, Koehler AP, Lawrence A, Sullivan TR, MacLennan JM, Maiden MCJ, Ladhani SN, Ramsay ME, Trotter C, Borrow R, Finn A, Kahler CM, Whelan J, Vadivelu K, Richmond P.

N Engl J Med. 2020 Jan 23;382(4):318-327.

B Part of It School Leaver protocol: an observational repeat cross-sectional study to assess the impact of a meningococcal serogroup B (4CMenB) vaccine programme on carriage of Neisseria meningitidis.

Marshall HS, McMillan M, Koehler A, Lawrence A, MacLennan J, Maiden M, Ramsay M, Ladhani SN, Trotter C, Borrow R, Finn A, Sullivan T, Richmond P, Kahler C, Whelan J, Vadivelu K.

BMJ Open. 2019 May 6;9(5):e027233.

Neisseria meningitidis serogroup B carriage by adolescents and young adults living in Milan, Italy: Prevalence of strains potentially covered by the presently available meningococcal B vaccines.

Terranova L, Principi N, Bianchini S, Di Pietro G, Umbrello G, Madini B, Esposito S.

Hum Vaccin Immunother. 2018 May 4;14(5):1070-1074. doi: 10.1080/21645515.2018.1450121. Epub 2018 May 9.

PMID: 29584565 Free PMC Article Select item 29565712

Impact of meningococcal vaccination on carriage and disease transmission: A review of the literature.

Balmer P, Burman C, Serra L, York LJ.

Hum Vaccin Immunother. 2018 May 4;14(5):1118-1130.

An unwanted guest: Neisseria meningitidis - carriage, risk for invasive disease and the impact of vaccination with insight on Italy incidence.

Gianchecchi E, Piccini G, Torelli A, Rappuoli R, Montomoli E.

Expert Rev Anti Infect Ther. 2017 Jul;15(7):689-701.

Author Response

The abbreviation for Neisseria has been added as have all the recent references suggested by the reviewer.

Reviewer 3 Report

The review itself was well written, the information is valid, but I found it to be more of an opinion article versus a review article that has a real synthesis of the peer reviewed work that is out there.

It did not offer any unknown path for future treatment of  Neisseria gonorrhoeae given its serious threat to public health. The review could have offered more depth with suggestions for the future that were more substantial and relevant.

Author Response

The article is a historical review of evidence. It does not purport to be a systematic review. It is true that the discussion does involve an interpretation of the historical evidence with regards to current concerns pertaining to AMR in gonococcus. If any specific arguments made in the discussion are regarded as too opinionated, these could be altered if necessary. If so please describe which sections of text require alteration?

Reviewer 4 Report

This review discusses the extent we should rely on antibiotics to reduce the prevalence of Neisseria gonorrhoeae especially in high risk communities. It is a very small review that has highlighted the findings of just few studies around the world. The author does mention the challenges of extrapolating the findings of studies with Neisseria meningitides to N. gonorrhoeae. Even though these are species from the same genus but very different pathogens. It is very difficult to follow this review written with limited review of literature. The narrative is frequently confusing and unclear what is being highlighted. Overall, I am unclear if this review adds anything to our current understanding of antibiotic use to reduce the prevalence of N. gonorrhoeae.

Author Response

The reviewer is thanked for their evaluation of this article. It is true that there are considerable differences between N. gonorrhoeae and N.meningitidis. We have acknowledged this in the following sections in the text:

Page 4, second paragraph: Although there are important differences in mode of transmission, preferred site of colonization, clinical presentation and host immune response between N. meningitidis and N. gonorrhoeae, there are also considerable similarities [11, 12]. Despite being the only 2 species in the Neisseria genus that are classified as strict human pathogens, the majority of both infections are asymptomatic and resolve spontaneously...

Page 7, second paragraph: There are also important biological differences between N. meningitidis and N. gonorrhoeae as well as differences between the mass administration of antibiotics during a meningococcal outbreak and the sustained high levels of antibiotic exposure in a PrEP cohort.

Reviewer 5 Report

This is an elegantly written review which makes a nice point concerning the possibility of selecting antibiotic resistant Neisseria gonorrhoeae in high prevalence populations which depend upon increased antibiotic consumption for infection control (e.g. PrEP cohorts). The author evaluates the literature on mass antibiotic treatment studies with N. meningitides to ascertain whether AMR emerged (it did, but not to antibiotics currently used to treat gonococcal infections –in fact there are no studies in meningococci that ascertain whether resistance to antibiotics would develop to such antimicrobials). The author also points out that there are also significant biological differences between the two Neisseria species.

Overall, the paper makes a cute point which should stimulate thinking. Has there been a good study which shows the effects of antibiotic exposure on commensal Neisseria sp? Why haven’t patients positive for N. gonorrhoeae also been tested for possible resistance development to N. meningitidis if they are carriers?

The sentence about horizontal gene transfer (lines120-122) requires some references. There is more than “some evidence” that commensal Neisseria sp play a role in eth development of resistance in N. gonorrhoeae, especially to third generation cephaloporins. This section/sentence should be expanded to reflect this.

How many mass treatment campaigns for N. gonorrhoeae have beenconducted (lines35-38). This could be expanded.

The author does not comment in the discussion about doxycycline pre-exposure prophylaxis and its possible effects on AMR.

Author Response

Reply to reviewer:

Overall, the paper makes a cute point which should stimulate thinking. Has there been a good study which shows the effects of antibiotic exposure on commensal Neisseria sp? Why haven’t patients positive for N. gonorrhoeae also been tested for possible resistance development to N. meningitidis if they are carriers?

Reply:

The following text has been added to expand on the evidence requested by the reviewer: (Page 7 last line to page 8)

Unsurprisingly, however, studies have found a link between antibiotic susceptibility of commensal Neisseria and antibiotic consumption [30]. A study from Japan found high proportions of circulating Neisseria subflava to have high miminum inhibitory concentrations for penicillin, cefixime, ciprofloxacin and tetracycline [30]. This was thought to be related to the high levels of the corresponding antimicrobial consumed in Japan [30, 31]. More direct evidence of this association comes from a study from Vietnam which found decreased cephalosporin susceptibility in commensal Neisseria to be related to recent cephalosporin consumption [32].

We could not find any studies that assess the effects of treatment for N. gonorrhoeae on cohabiting N. meningitidis.

The sentence about horizontal gene transfer (lines120-122) requires some references. There is more than “some evidence” that commensal Neisseria sp play a role in eth development of resistance in N. gonorrhoeae, especially to third generation cephaloporins. This section/sentence should be expanded to reflect this.

Reply:

This section has been strengthened by the inclusion of the following text and 3 new references: (Page 7, last paragraph)

There is increasing evidence that horizontal gene transfer plays an important role in the genesis of AMR in N. meningitidis and even more so in N. gonorrhoeae. For example, it has been established that transformation from commensal pharyngeal Neisseria spp. played an important role in N. gonorrhoeae’s acquisition of resistance to extended spectrum cephalosporins [30-32]. The acquisition of AMR via commensals can operate over much longer periods than direct selection during antibiotic therapy, as the commensals (and their resistance conferring genes) persist for longer periods than N. gonorrhoeae.

How many mass treatment campaigns for N. gonorrhoeae have beenconducted (lines35-38). This could be expanded.

Reply:

This section has been expanded to include references to the four most relevant trials of mass STI treatment that evaluated the effect on N. gonorrhoeae incidence/prevalence. Three new references have been added to this section: (Page 3, last paragraph)

There have been few mass treatment trials of N. gonorrhoeae. These studies have typically found that mass treatment has no effect [5-7], or only a temporary effect on gonorrhoea incidence/prevalence [8, 9]. Only one of these studies evaluated the effect on AMR. Although this study found a temporal association between mass treatment and the emergence of gonococcal AMR, its contemporary relevance is reduced by the fact that it was conducted using penicillin in the 1960s [8, 9].

The author does not comment in the discussion about doxycycline pre-exposure prophylaxis and its possible effects on AMR.

Reply:

The following has been added to the discussion to address this point: (Page 8, second paragraph)

Two trials have been conducted to assess if doxycycline pre- and post-exposure could reduce the incidence of bacterial STIs including gonorrhoea [31, 32]. Both studies found evidence of moderate reductions in chlamydia and syphilis incidence but not gonorrhoea. The effect on AMR of pathogens and commensal organisms was not assessed in these studies.

Round 2

Reviewer 3 Report

No further comments.

Reviewer 4 Report

The authors had modified the manuscript and made it suitable for publication.